# Loss-of-Function Variants in *SUPT5H* as Modifying Factors in Beta-Thalassemia

**DOI:** 10.3390/ijms25168928

**Published:** 2024-08-16

**Authors:** Cornelis L. Harteveld, Ahlem Achour, Nik Fatma Fairuz Mohd Hasan, Jelmer Legebeke, Sandra J. G. Arkesteijn, Jeanet ter Huurne, Maaike Verschuren, Sharda Bhagwandien-Bisoen, Rianne Schaap, Linda Vijfhuizen, Hakima el Idrissi, Christian Babbs, Douglas R. Higgs, Tamara T. Koopmann, Christina Vrettou, Joanne Traeger-Synodinos, Frank Baas

**Affiliations:** 1Department of Clinical Genetics/LDGA, Leiden University Medical Center, P.O. Box 9600, 2333 ZC Leiden, The Netherlands; 2Department of Congenital and Hereditary Diseases, Charles Nicolle Hospital, Tunis 3000, Tunisia; 3Department of Pathology, Hospital Raja Perempuan Zainab II, Kota Bharu 15400, Malaysia; 4Radcliffe Department of Medicine, Medical Sciences Division, University of Oxford, Oxford OX3 9DU, UK; 5Laboratory of Medical Genetics, National and Kapodistrian University of Athens, St. Sophia’s Children’s Hospital, 115 27 Athens, Greece

**Keywords:** hemoglobin, *SUPT5H*, beta-thalassemia, hematology, modifying factor, molecular diagnosis, β-thalassemia intermedia

## Abstract

It is well known that modifiers play a role in ameliorating or exacerbating disease phenotypes in patients and carriers of recessively inherited disorders such as sickle cell disease and thalassemia. Here, we give an overview of the literature concerning a recently described association in carriers of *SUPT5H* Loss-of-Function variants with a beta-thalassemia-like phenotype including the characteristic elevated levels of HbA_2_. That *SUPT5H* acts as modifier in beta-thalassemia carriers became evident from three reported cases in whom combined heterozygosity of *SUPT5H* and *HBB* gene variants was observed to resemble a mild beta-thalassemia intermedia phenotype. The different *SUPT5H* variants and hematologic parameters reported are collected and reviewed to provide insight into the possible effects on hematologic expression, as well as potential disease mechanisms in carriers and patients.

## 1. Introduction

Hemoglobinopathies are the most common monogenic disorders in the world, with an ever-increasing global disease burden each year [1,2]. The thalassemias are characterized by reduced synthesis of the globin chains of hemoglobin, specifically beta-globin chains in beta-thalassemia. As most hemoglobinopathies show recessive inheritance, carriers are usually clinically silent, although they can be identified based on hematological features including microcytic hypochromic erythrocytes and elevated levels of the minor adult hemoglobin known as HbA_2_. Regular hemoglobinopathy diagnostics as well as pre-marital, pre-conceptional and/or antenatal thalassemia screening programs occasionally identify individuals with microcytic hypochromic parameters and elevated HbA_2_ in whom no beta-thalassemia variants in the *HBB* gene can be identified [3,4]. In very rare cases, haplotype analysis may reveal a pattern of inheritance that does not segregate with the locus of the *HBB* gene located within the so-called beta-globin gene cluster (chromosome 11 p15), so-called unlinked beta-thalassemia [3]. This can be caused by other genes in the human genome involved in regulating beta-globin gene expression. Several erythroid-specific genes have been reported to carry variants causing elevation of the percentage of HbA_2_, such as *GTF2E2*, *GATA1*, *ASH1L* and *KLF1* [5,6,7,8,9], although not necessarily with beta-thalassemia traits like red cell indices. 

Recently, Whole Exome Sequencing (WES) identified Loss-of-Function (LoF) variants in *SUPT5H* (NM_001111020.3) in several members of two independent Dutch families showing microcytic hypochromia and elevated HbA_2_ but no beta-thalassemia variants in the *HBB* genes [3]. The discovery of splicing variants (including both donor- and acceptor variants) in *SUPT5H* in these two independent families and subsequent RNA sequencing confirmed an altered splice pattern leading to intron retention and a premature stop codon. 

Additional LoF variants in *SUPT5H* were found in several other individuals of Dutch, French and Italian ancestry, all of whom presented hematological characteristics consistent with beta-thalassemia traits but were negative for variants in the *HBB* genes [3,10,11]. Three individuals from two independent Greek families expressing a non-transfusion-dependent beta-thalassemia intermedia phenotype were characterized to have co-inherited a known Mediterranean beta-thalassemia variant and an additional variant in *SUPT5H*. These findings indicated that *SUPT5H* may play a role in regulating beta-globin gene expression. Elevated HbA_2_ in the absence of a beta-thalassemia variant in the *HBB* gene is one of the key hematological findings that may suggest the presence of a variant in *SUPT5H*. More cases have been reported with unknown *SUPT5H* variants expressing beta-thalassemia traits after the first publication by Achour et al. in 2020 [3]. 

## 2. *SUPT5H* Gene and hSpt5 Protein: Structure and Function

Promoter proximal pausing is an important regulatory step in eukaryotic transcription catalyzed by the enzyme RNA polymerase II (RNA Pol II). Participation in the general transcription factor DSIF is required for normal RNA synthesis. Human DSIF is a heterodimer composed of two subunits, hSpt4 and hSpt5. From the N-terminus, the hSpt5 protein contains an acidic domain, a region that is homologous to the bacterial transcription factor NusG (NGN domain), which interacts with Spt4h, followed by five Kyrpides–Ouzounis–Woese (KOW) domains, two C-terminal repeat regions and another two C-terminal KOW domains (Figure 1). The two C-terminal repeat regions (CTR-1 and CTR-2) can be phosphorylated by the positive transcription elongation factor pTEFb activating RNA Pol II elongation [12]. The C-terminal region of hSpt5 includes the tandem KOW domain, designated KOW6 and 7 in Figure 1, which is near the exiting RNA and might play a role in recruiting factors for RNA capping and in 3′RNA processing [13,14]. Zuber et al. (2018) [12] demonstrated a tight domain interaction between KOW4, KOW6 and KOW7, a typical beta-barrel fold that possibly enlarges the structure of the basic KOW fold compared to other KOW domains of hSpt5. This extended structure might be of importance to present a larger binding surface for additional molecular interactions. The KOW6-7 domains were shown to interact exclusively with protein factors necessary for RNA elongation and/or processing during transcription termination. 

If an *SUPT5H* LoF variant leads to a truncated Spt5 protein, parts of the C-terminal domains are predicted to be missing. Depending on the domains that are missing, important regions include the NGN region (critical for Spt4 dimerization), the CTR-1 and 2 regions (sensitive to phosphorylation and, therefore, activation of the DSIF complex) and the KOW6-7 domains (necessary for binding protein factors involved in RNA elongation, processing and termination). It is currently unclear whether a non-functional truncated protein product is formed or whether the transcript is degraded by Nonsense-Mediated Decay (NMD). In any case, there is no apparent correlation between the location of the premature stop position and the hematologic indices observed amongst carriers of the diverse *SUPT5H* variants. For example, heterozygotes for the nonsense variants c.193C>T and c.142G>T, both predicted to lead to a truncated protein missing all functional domains, present with hematologic parameters comparable to those in heterozygotes for the most terminal c.3032_3033delTG frameshift variant. The latter codes for an almost intact Spt5 protein in which only the last KOW domain and the C-terminus are missing [4,15]. These findings could support the hypothesis of NMD; however, more studies are necessary to shed light on how the reduction of Spt5 levels affects the beta-globin LCR recruitment of transcription factors to eventually reduce the expression of the *HBB* genes.

## 3. Molecular *SUPT5H* Variants Reported

A total of 28 *SUPT5H* variants have been reported to date. In the first report describing *SUPT5H* as a gene associated with a beta-thalassemia trait, eight different variants were observed [3]. Subsequent papers have reported an additional 20 variants. Heterozygotes may present either near-to-normal hematology or microcytic hypochromic parameters, but all were observed to have elevated HbA_2_ levels comparable to those in beta-thalassemia traits [4,10,11,15,16,17]. To date, there are no further reports of cases with double heterozygosity for variants in *SUPT5H* and *HBB*, and the only reason *SUPT5H* variants were suspected was due to elevated HbA_2_ levels with or without microcytic hypochromic indices in the absence of variants in the *HBB* gene. 

The *SUPT5H* variants presently known are listed in Table 1 and shown in Figure 1. The majority are either nonsense or frameshift variants leading to a premature stop codon. Of four variants reported which alter the splice consensus sequence, one splice donor variant and one splice acceptor variant (c.458+1G>A and c.2259-3C>A, respectively) have been studied at the RNA level, demonstrating intron retention and an altered reading frame leading to a premature stop codon. Of the other splicing variants, two have not been examined at the RNA level, but in silico prediction software suggests an effect on splicing. The in silico prediction of two missense variants, *SUPT5H*:c. 2245C>T, p.(Arg749Trp) and c.2507A>G, p.(Tyr836Cys), also indicates alternative splicing, a subsequent altered reading frame and a premature stop codon. All variants, therefore, are consistent with the synthesis of a truncated protein, missing the 3′carboxy-terminus. The only exception is the variant *SUPT5H*: c.3034T>A, p.(Cys1012Ser), a missense variant in the penultimate exon of the gene without a clear in silico prediction in favor of alternative splicing. 

As almost the entire spectrum of *SUPT5H* variants in individuals expressing an unlinked beta-thalassemia phenotype from different geographic areas lead to an LoF, this suggests that an intact C-terminus is essential for the correct function of the Spt5 protein. The haplo-insufficiency of Spt5, or a truncated Spt5 which misses the carboxy-terminus, is apparently the mechanism which gives rise to the typical phenotype of beta-thalassemia traits with the characteristically elevated levels of HbA_2_. 

## 4. In Vitro Studies of *SUPT5H* LoF

The *SUPT5H* gene codes for human Spt5, a protein known to play a role in controlling the release of promoter-proximal pausing of RNA Pol II, which is crucial for gene regulation and cellular differentiation. That RNA pol II pausing plays a role in modulating hematopoietic stem cell emergence was already shown previously in zebrafish [18,19]. During erythropoiesis, cell cycle genes are largely paused as cells transition from progenitors to precursors. RNA Pol II pauses ~30 to 50 bp downstream of promoters and its release into productive elongation is a highly regulated process [20,21,22,23]. One of the factors involved in the convergence from paused to released RNA Pol II is the DRB (5,6,-dichloro-1-beta-D-ribofuranosylbenzimidazole) sensitivity-inducing factor (DSIF complex), composed of Spt5 and Spt4. As Spt5 is an essential and highly conserved elongation factor, depletion often leads to cell death and its role in cellular differentiation is therefore difficult to study [24,25,26,27,28]. The identification of LoF mutations in *SUPT5H* in patients with a beta-thalassemia trait-like phenotype in the absence of *HBB* mutations created a unique opportunity to study RNA Pol II pausing [11]. By perturbing *SUPT5H* in human hematopoietic stem and progenitor cells (HSPCs) using CRISPR-Cas9 to generate a heterozygous population of cells with a 50% reduced level of Spt5 protein levels, it was possible to study the effect on globin expression and cell differentiation. 

Martell et al. (2023) [11] observed that RNA Pol II pause release, which facilitates productive elongation, was hindered in human hematopoiesis with disrupted *SUPT5H* expression. As these cells began transitioning from progenitors to precursors, both cell cycle kinetics and the onset of erythroid gene expression programs were delayed, although cells still underwent terminal differentiation. This suggests that RNA Pol II pausing plays a role in the regulation of cell cycle progression during erythroid differentiation as cells begin transitioning from progenitors into more mature erythroid precursors. The authors also examined the level of non-coding enhancer-derived RNAs (eRNAs) in the regulatory regions of both the beta- and alpha-globin gene clusters, as eRNA transcription is considered a reliable marker of enhancer activity [29,30]. The reduced levels of eRNA within the beta-globin Locus Control Region (beta-LCR) observed in *SUPT5H*-edited cells suggests that this may contribute to the reduced *HBB* expression in carriers of *SUPT5H* LoF variants. *HBB* expression is more susceptible than *HBA* expression to transcription perturbations, particularly at a later stage of erythropoiesis, when there is a higher demand for globin expression. It is not clear why *HBB* expression is affected more than *HBA* expression, but as the modes of regulation of gene expression in the beta- and alpha-globin gene clusters are different, it is possible that the effect of perturbation of RNA Pol II pausing may also be different. As Martell et al. (2023) [11] clearly state, patients harboring these mutations are relatively healthy, exhibiting subtle but consistent phenotypes characterized by mild or no anemia but clear signs of beta-/alpha-globin chain imbalance. This was confirmed by the imbalance in globin chain-synthesis results in one of the *SUPT5H* carriers reported by Achour et al. (2020) [3] and supports a down-regulation of *HBB* expression but not of *HBA1/2* expression, or at least not to the same extent. The fact that *HBB* expression is regulated through the recruitment of Spt5 via the beta-LCR suggests that this may be an explanation for the reduced *HBB* expression observed in carriers of *SUPT5H* variants. In contrast, *HBA* enhancer sites were largely unaffected in *SUPT5H*-edited cells, with the exception of one site which was significantly up-regulated, which suggests that LoF variants can have differential effects on enhancer transcription across these different co-expressed genes [31].

## 5. Hematologic Phenotype of Carriers of LoF *SUPT5H* Variants

How do carriers of *SUPT5H* variants compare to carriers of known beta-thalassemia variants at the hematological level? The mean and range of hemoglobin concentration (Hb g/L), Mean Cellular Volume (MCV fL), Mean Corpuscular Concentration of Hemoglobin (MCH pg) and HbA_2_ values were collected from 47 *SUPT5H* heterozygotes reported in the literature (Appendix A) and compared to the mean values and ranges for Hb, MCV, MCH and HbA_2_ levels for the normal [32] and beta-thalassemia carriers [1] as known from the literature. For *SUPT5H*/*HBB* double heterozygotes, the numbers were too limited to determine a mean or range; they are depicted as values in the graph (Figure 2). 

The comparison between the four groups, consisting of normal, beta-thalassemia carriers, *SUPT5H* carriers and double heterozygotes for *SUPT5H* and *HBB* variants, shows a decrease in Hb, MCV and MCH values between the normal and the beta-thalassemia carriers (Figure 3a–c). The comparison of hematologic indices (Hb, MCV and MCH) between the *SUPT5H* carriers, the normal range and the beta-thalassemia carrier range demonstrates that *SUPT5H* carriers show anemia due to a decrease in hemoglobin concentration equivalent to that seen in beta-thalassemia carriers. As far as MCV and MCH are concerned, the mean value for the *SUPT5H* carriers is between the normal and beta-thalassemia carriers, while the range overlaps between normal and beta-thalassemia carriers; this is a finding consistent with the relatively mild effect on the bi-allelic reduction in beta-gene expression of the *SUPT5H* carriers. Only three cases of double heterozygosity for an *SUPT5H* variant and a beta-thalassemia determinant in the *HBB* gene have been reported, showing Hb levels falling within the lower range of beta-thalassemia carriers. The MCV and MCH values seem to be below average compared to the beta-thalassemia group. Even though the number of double heterozygotes is not enough to calculate reliable statistical differences, these findings suggest that there is a modifying effect induced by *SUPT5H* haplo-insufficiency that down-regulates the beta-globin gene expression in beta-thalassemia heterozygotes. As the *SUPT5H* protein Spt5 acts transiently, it seems most likely that *HBB* expression on both alleles is down-regulated, enhancing the imbalance between alpha- and beta-globin polypeptide synthesis. Considering that carriers of *SUPT5H* variants have a hematological phenotype comparable to that of beta-thalassemia carriers, the expected phenotypic expression of double heterozygotes is more severe, which seems to be confirmed by the hematologic data. 

No significant differences are seen in the percentage of HbA_2_ in beta and *SUPT5H* carriers (Figure 3d). However, HbA_2_ levels were significantly elevated in double heterozygous *SUPT5H*/beta-thalassemia carriers. As there are only three reported cases of combined beta-thalassemia and *SUPT5H* LoF variants, it is difficult to determine if this is due to the specific feature of the beta-thalassemia variant or the effect of the interaction of the two variant alleles. Two of the cases described to date are double heterozygotes for *HBB*:c.118C>T and *SUPT5H*:c.1374+2T>C (with HbA_2_ levels of 8.5% and 11.1%, respectively). Heterozygotes for *HBB*:c.118C>T usually present with relatively lower HbA_2_ levels of 4.89 ± 0.84%. The third case is a double heterozygote for *HBB*:c.92+1G>A and *SUPT5H*:c.1741_1744dup (HbA_2_ 12.4%). The heterozygote *HBB*:c.92+1G>A presents with a higher HbA_2_ of approximately 9–10%. The extra-ordinary elevation in HbA_2_ percentage seen in two double heterozygotes, *HBB*:c.118C>T and *SUPT5H*:c.1374+2T>C, cannot be attributed to the *HBB* variant. In beta-thalassemia carriers expressing a more severe clinical phenotype than expected from beta-thalassemia traits, such elevated HbA_2_ percentages might be an indication for a co-inherited *SUPT5H* variant.

## 6. Conclusions

LoF variants in *SUPT5H* are associated with a beta-thalassemia-like phenotype in so-called unlinked beta-thalassemia cases with typically elevated HbA_2_ and, in most cases, microcytic hypochromic erythrocyte parameters. *SUPT5H* is a modifying factor which may play a role in beta-thalassemia carriers who express a more severe phenotype than is seen in simple beta-thalassemia heterozygotes. However, examples from the literature are apparently scarce, with only three reported cases to date.

Recent studies have demonstrated that diminished *SUPT5H* expression levels can induce a stage-specific delay in erythroid differentiation [11]. In spite of extensive analysis of the Spt5 mutations in human HSPCs, we do not fully understand the mechanism with which Spt5 acts on beta-globin gene expression. It is also not clear why Spt5, which seems to play such an important role in RNase II polymerase pausing and elongation, has no detectable effect on alpha-globin gene expression in these cell lines, nor gives rise to a more severe pathological phenotype in carriers with abnormal development in other tissues.

The comparison of red cell indices amongst different groups of reference samples, beta-thalassemia carriers and carriers of *SUPT5H* variants and compound heterozygotes provided more insight into the clinical phenotype of reduced *SUPT5H* expression. The elevated levels of HbA_2_ in the absence of a beta-thalassemia trait causing variants in the *HBB* gene is one of the most suggestive elements to identify variants in the *SUPT5H* gene. Carriers of *SUPT5H* LoF variants express a mild beta-thalassemia trait, while double heterozygotes for *SUPT5H* and a beta-thalassemia variant present with a mild beta-thalassemia intermedia phenotype. For the latter, elevated HbA_2_ levels of 10.7% (SD = 2.0%) potentially implicate a diagnosis of combined *SUPT5H*/*HBB* in beta-thalassemia carriers expressing beta-thalassemia intermedia. However, little is known about the clinical impact of double heterozygosity as detailed information about increased transfusion need, splenomegaly, manifestations of extramedullary hematopoiesis, etc., is missing, with only three cases reported in the literature. The clinical severity may also impact genetic counseling when couples are identified as being carriers of an *SUPT5H* LoF variant and a regular beta-thalassemia trait variant in *HBB*, respectively. As only a few cases are known from the literature, it remains difficult to predict the severity of the beta-thalassemia intermedia phenotype in an affected double heterozygous child. If both parents are carriers of *SUPT5H* LoF variants, 25% of pregnancies will not lead to a viable embryo as the total lack of *SUPT5H* expression is likely incompatible with life, with hSpt5 playing such an essential role in the biologically essential mechanism of RNA Pol II pausing and elongation.

As the modifying effect of *SUPT5H* LoF is relatively subtle, it might be underdiagnosed amongst beta-thalassemia carriers with a more severe clinical expression. The extraordinarily elevated HbA_2_ levels may be a biomarker for the presence of an additional *SUPT5H* variant in cases in which a single *HBB* variant does not explain the clinical phenotype. Next Generation Sequencing (NGS) to investigate the *SUPT5H* gene would be the next diagnostic step or, if not available, contacting local, more specialized clinical genetic laboratories for further testing.

More studies are necessary to investigate the mechanism underlying reduced beta-gene expression in carriers of *SUPT5H* variants. The hemoglobinopathy network INHERENT (URL https://www.inherentnetwork.org (accessed on 14 August 2024)) is instrumental in investigating this topic further. International collaboration between diagnostic laboratory scientists and hematologists is essential in identifying cases where the clinical severity is not fully explained by the genotype or the elevated HbA_2_ is not explained by variants in the *HBB* gene.

## Figures and Tables

**Figure 1 ijms-25-08928-f001:**
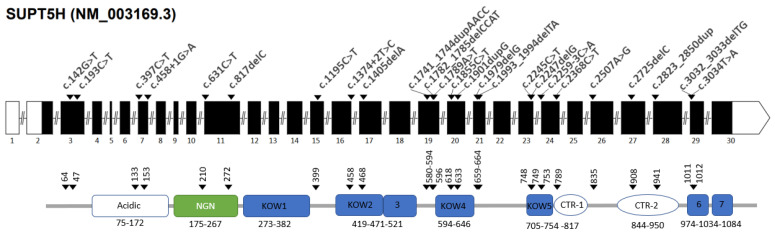
Schematic presentation of the *SUPT5H* gene. The white boxes are the untranslated regions, and the black boxes are the coding exons. Arrows indicate the position and HGVS annotation of the variants. The protein structure is shown below from left (N-terminus) to right (C-terminus). The functional domains are the acidic region, the NGN domain which interacts with Spt4, the five KOW domains (Kyrpides–Ouzounis–Woese domains), two C-terminal repeat domains (CTR-1 and CTR-2) which can be phosphorylated and, finally, two KOW domains (KOW6 and 7). The numbers below the domains indicate the amino acid (a.a.) positions in the protein chain and the arrows correspond with the arrows in the upper graph, indicating the a.a. position of each variant.

**Figure 2 ijms-25-08928-f002:**
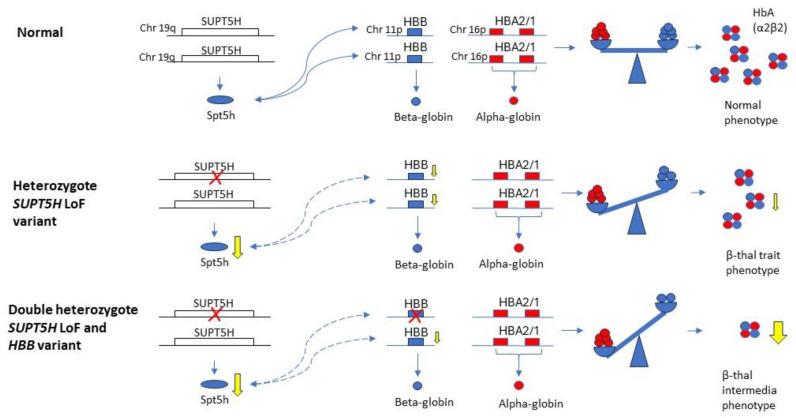
Schematic presentation of the effect of reduced *SUPT5H* expression in carriers and in double heterozygous beta-thalassemia trait carriers. The *SUPT5H* gene is indicated as an open box, *HBB* and *HBA1/2* genes as blue and red boxes respectively. The proteins are indicated below the genes as oval (Spt5h) and circles (alpha-/beta-globin). The tetramer represents hemoglobin HbA, the yellow arrows at the genes indicate the reduced levels of expression and reduction of protein synthesis, the curved lines indicate a suspected influence of reduced Spt5h synthesis on the *HBB* expression, for which the exact mechanism is unknown.

**Figure 3 ijms-25-08928-f003:**
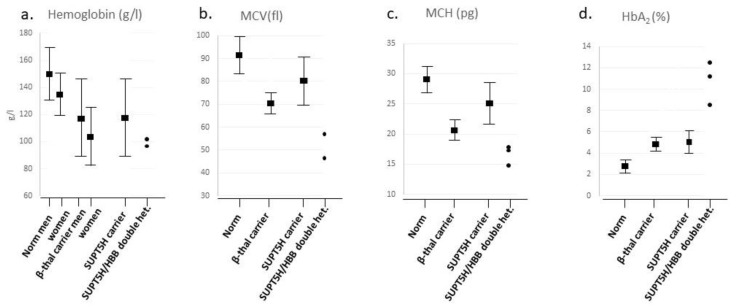
(**a**). Mean and 95% range of red cell indices for reference values from Dacie and Lewis, Practical Haematology 12th ed. [32], β-thalassemia carriers from Weatherall and Clegg, The Thalassemia Syndromes 4th ed. (1), *SUPT5H* carriers and double heterozygous carriers of *SUPT5H* and *HBB* variants (Appendix A). (**a**). Hb (g/L) mean ± 2SD for reference group of men is 150 ± 20 g/L and that for women is 135 ± 15 g/L; Hb (g/L) mean ± 2SD for *SUPT5H* carriers is 119.5 g/L (91–149 g/L), which overlaps largely with the Hb of β-thalassemia carriers (men, 118 ± 15 g/L, and women, 108 ± 9.0 g/L). Mean Hb level of double heterozygote β-thalassemia and *SUPT5H* carriers is 98 g/L (SD = 2.89). (**b**). Mean MCV of reference men and women is 92 ± 9 femtoliter; the MCV of *SUPT5H* carriers is 79 femtoliter (69–91); the MCV of β-thalassemia carriers is 70.5 + 4.2 fl. The mean MCV of double heterozygote β-thalassemia and *SUPT5H* carriers is 53.3 femtoliter (SD = 6.35). (**c**). Mean MCH of reference is 29.5 ± 2.5 picogram, that of β-thalassemia carriers is 21.5 ± 1.3 picogram and that of *SUPT5H* carriers is 25.60 picogram (21.6–27.4). Mean MCH of double heterozygote β-thalassemia and *SUPT5H* carriers is 16.6 (SD = 1.42). (**d**). Mean HbA_2_ percentage for reference group is 2.2–3.5, mean HbA_2_% of *SUPT5H* carriers is 5.2% (3.6–6.4), and mean HbA_2_% of β carriers is 4.9 ± 0.5%. Mean of HbA_2_% for double heterozygote β-thalassemia and *SUPT5H* carriers is 10.67% (SD = 1.99).

**Table 1 ijms-25-08928-t001:** Variants in *SUPT5H* (NM_001111020.3). * Delta score [0–1.0] generated by SpliceAI # GnomAD v4.0.0.

Nr.	HGVS Annotation	Molecular Effect	In Silico Prediction *	Description	Etnic Origin	GnomAD Allele Frequency #	Reference
1	c.142G>T	p.Glu48*		nonsense	Chinese		[4]
2	c.193C>T	p.Arg65*		nonsense	Chinese	7.15 × 10^−7^	[15]
3	c.397C>T	p.Arg133*		nonsense	Chinese	3.05 × 10^−6^	[17]
4	c.458+1G>A		1.00	Donor loss	Dutch		[3]
5	c.631C>T	p.Gln211*		nonsense	Chinese		[4]
6	c.817delC	p.Leu273*		nonsense	French		[3]
7	c.1195C>T	p.Gln399*		nonsense	Chinese		[16]
8	c.1374+2T>C		0.71	Donor loss	Greek		[3]
9	c.1405delA	p.Arg469Glufs*10		Frameshift	Chinese		[4]
10	c.1741_1744dupAACC	p.Arg582Glnfs*21		Frameshift	Greek		[3]
11	c.1782_1785delCCAT	p.Ile594Metfs*2		Frameshift	Chinese?		[3]
12	c.1789A>T	p.Leu597*		nonsense	?		[11]
13	c.1855C>T	p.Arg619*		nonsense	?		[4]
14	c.1901dupG	p.Met635Hisfs*19		Frameshift	Chinese		[4]
15	c.1979delG	p.Gly660Valfs*6		Frameshift	Dutch		[3]
16	c.1993_1994delTA	p.Met665Glufs*19		Frameshift	?		[11]
17	c.2245C>T	p.Arg749Trp	0.69	Donor gain	?	2.40 × 10^−6^	[11]
18	c.2247delG	p.Leu750Serfs*11		Frameshift	?		[11]
19	c.2259-3C>A		0.09	-	Dutch		[3]
20	c.2368C>T	p.Gln790*		nonsense	French		[10]
21	c.2507A>G	p.Tyr836Cys		Alternative splicing	?		[11]
22	c.2725delC	p.Gln909Argfs*45		Frameshift	Dutch		[3]
23	c.2823_2850dup	p.Ser951*		nonsense	Chinese		[4]
24	c.3032_3033delTG	p.Met1011Metfs*9		Frameshift	Chinese		[17]
25	c.3034T>A	p.Cys1012Ser	0.02	-	?		[11]
26		EXON 21 -2 A>G		Splice acceptor	?		[11]
27		p.Glu455Aspfs*23		Frameshift	?		[11]
28		del(chr19:39936531-40030719)		?	Chinese		[4]

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
