# Peer review of "Loss-of-Function Variants in SUPT5H as Modifying Factors in Beta-Thalassemia"

_ijms, 2024, doi:10.3390/ijms25168928_

Round 1

Reviewer 1 Report

Comments and Suggestions for Authors

In this review, the authors summarize the available evidence on the role of SUPT5H LoF variants as cause of beta-thalassemia-trait-like hematological findings and as an important genetic modifier in beta-thalassemia trait.

This information is part of a novel, very interesting and (potentially) clinically very relevant field for the diagnostics and the clinical management of individuals with unexpected and unexplained RBC changes in the blood count.

The manuscript is overall well written and included all relevant information on the topic. There are no critical remarks to this work. However, some adjustments could ameliorate the legibility of the text and some points deserve to be put more in evidence.

Remarks:

-       Structure of the manuscript: Suggestions for the sequence of the paragraphs and for their title:

·         Introduction (this part is clear and comprises all relevant points regarding the background of the review)

·         SUPT5H gene and hSpt5 protein: structure and function. The reviewer suggests starting with a description of both the gene and the protein

·         Molecular SUPT5H variants reported

·         In-vitro studies of SUPT5H loss-of-function (instead of “In-vitro studies of hSpt5 haplo-insufficiency”)

·         Hematologic phenotype of carriers of loss-of function SUPTH5 variants (instead of “Hematology in carriers and patients”)

·         Conclusions

 -       Significance of elevated HbA2: an elevated HbA2 in the absence of a beta-thal trait is one of the key hematological findings that should suggest the presence of a SUPT5H variant (see in particular Lou et al, REF 4). It would be important to describe in more detail the changes of HbA2 in the text and stress this point a little more as one of the most suggestive hematological values for the presence of a SUPT5H variant. For example, a sentence in the Introduction and a comment in the section “Hematology in carriers and patients” could be added.

 -       It is important to underline that the available data focused on the hematological phenotype of carriers of the SUPT5H variants. The actual clinical impact of the variants (increased need for transfusion, splenomegaly, manifestations of extramedullary hematopoiesis etc) is little known except for the few cases with both a beta-thal trait and a SUPT5H LoF variant. The clinical features of these cases, however, were not described in detail in the corresponding publications. Therefore, I suggest omitting the term “clinical” (e.g. in the abstract) and “patients” (in “Hematology in carriers and patients”) and limit the description and comments to the hematological phenotypic manifestations.

 -       As a suggestion, the following points should be also briefly discussed or mentioned:

·         the impact of SUPT5H variants on the expression of alpha-globin chains (what is known about that and what is not)

·         the relevance of testing persons with unexplained “beta-thal minor” phenotype for SUPT5H variants for genetic counselling (e.g. a sentence in the section “Hematology in carriers and patients” and/or in the Conclusions.)

·         The need to contact specialized laboratories for this testing

 -       The description of structure and function is exhaustive and understandable, but maybe a bit difficult to follow. Would a cartoon be of help?

 -       Abbreviations: these should be used consistently through the manuscript, e.g. RNA Pol II, LoF

Comments on the Quality of English Language

As far as possible for the reviewer to judge, there are only a few minor language issues.

Reviewer 2 Report

Comments and Suggestions for Authors

This is a very well written and interesting review on a genetic modifier of the thalassemia trait which is less known than others. My major concern is that the text mentions data presented in Table 1 but I did not find any Table 1 in the body of the manuscript. I have also a slightly minor concern.  The text mentions that the levels of HBA2 are also increased by variants in GTF2E2, GATA1, ASH1L and KLF1. A table with a side by side comparison on how variants in GTF2E2, GATA1, ASH1L and KLF1 affect HBA2 expression with respect to the SUPT5H  variants with and without a thalassemic trait would increase the completeness of the report.  
